# New Insights about CuO Nanoparticles from Inelastic Neutron Scattering

**DOI:** 10.3390/nano9030312

**Published:** 2019-02-26

**Authors:** Elinor C. Spencer, Alexander I. Kolesnikov, Brian F. Woodfield, Nancy L. Ross

**Affiliations:** 1Department of Geosciences, Virginia Polytechnic Institute and State University, Blacksburg, VA 24061, USA; espence@gmail.com; 2Neutron Scattering Division, Oak Ridge National Laboratory, Oak Ridge, TN 37831, USA; kolesnikovai@ornl.gov; 3Department of Chemistry and Biochemistry, Brigham Young University, Provo, UT 84602, USA; Brian_Woodfield@byu.edu

**Keywords:** copper oxide, nanoparticles, neutron scattering, magnetism

## Abstract

Inelastic Neutron Scattering (INS) spectroscopy has provided a unique insight into the magnetodymanics of nanoscale copper (II) oxide (CuO). We present evidence for the propagation of magnons in the directions of the ordering vectors of both the commensurate and helically modulated incommensurate antiferromagnetic phases of CuO. The temperature dependency of the magnon spin-wave intensity (in the accessible energy-range of the experiment) conforms to the Bose population of states at low temperatures (*T* ≤ 100 K), as expected for bosons, then intensity significantly increases, with maximum at about 225 K (close to *T*_N_), and decreases at higher temperatures. The obtained results can be related to gradual softening of the dispersion curves of magnon spin-waves and decreasing the spin gap with temperature approaching *T*_N_ on heating, and slow dissipation of the short-range dynamic spin correlations at higher temperatures. However, the intensity of the magnon signal was found to be particle size dependent, and increases with decreasing particle size. This “reverse size effect” is believed to be related to either creation of single-domain particles at the nanoscale, or “superferromagnetism effect” and the formation of collective particle states.

## 1. Introduction

Magnetism exhibited by nanoscale materials (so-called “nanomagnetism”) is not a fully understood phenomenon, and despite evidence to suggest that ferromagnetism is a universal feature of nanoscale metal oxides [1], research in this field is still in its infancy. The magnetic behavior of nanoparticles is well known to deviate significantly from bulk materials due to pronounced surface effects and/or the creation of single-domain particles [2,3,4,5,6]. Oxygen vacancies [7,8], surface tension [9,10], uncompensated surface spins and the exchange interactions between these spins and those within the core of the particle [3,11] can all significantly alter the magnetic ordering and transition temperatures of magnetic materials.

Despite the known toxicity of copper (II) oxide (cupric oxide, CuO) nanoparticles [12,13], they are finding application in numerous commercially viable arenas due to their demonstrated efficacy as catalysis for complex chemical reactions [14,15,16], and as biological mimetics for the sensing of small molecules [17,18]. However, the magnetic properties of CuO nanoparticles are currently underexploited, certainly in comparison with other magnetic transition metal oxides such as the iron oxides Fe_3_O_4_, α-Fe_2_O_3_, and γ-Fe_2_O_3_. A possible explanation for this is that the often complex magnetic behavior of the copper oxides is perhaps one of the least understood, particularly at the nanoscale. However, recent evidence of multiferroicity and electromagnons in CuO has provided impetus for renewed interest in this simple oxide [19,20,21]. Moreover, while most simple monoxides containing 3*d* transition metals adopt the cubic rocksalt structure, CuO adopts a monoclinic (space group C2/*c*) structure [22,23,24]. In the structure, Cu is surrounded by four O atoms in a square planar configuration, which form ribbons of edge-sharing chains running along the [101] and [10$\bar{1}$] directions [22,23,24].

This unique structure gives rise to the unusual magnetic properties of CuO [5,25]. The majority of magnetic oxides exist in a completely disordered paramagnetic state above their individual Néel temperatures (*T*_N_) and convert completely to give 3D antiferromagnetic states below these temperatures. However, bulk CuO behaves very differently by undergoing two magnetic phase transitions. Above the first *T*_N_ at approximately (ca.) 230 K (*T*_N1_) antiferromagnetic dynamical spin correlations along the crystallographic [10$\bar{1}$] direction is still maintained up to high temperatures, but there is no long-range spin order and the material is in paramagnetic state. Below *T*_N1_, a 3D incommensurate state is created in which the spins are ordered in a non-collinear helical arrangement. The second magnetic transition at ca. 213 K (*T*_N2_) induces the spins to adopt a collinear configuration and a fully commensurate 3D antiferromagnetic structure results below this temperature [26,27].

Importantly, *T*_N_ is strongly suppressed in nanoparticulate CuO, is ca. 30 K for 5 nm particles, and as low as 13 K for particles 2–3 nm in size [22]. This is believed to be a consequence of the dependency of *T*_N_ on the strength and number of superexchange interactions, which are drastically reduced/weakened at the nanoscale relative to the bulk oxide; this is because of a significant increase in the number of uncompensated surface spins and a reduction in the number of exchange pathways (via oxygen orbitals) due to an increase in the number under-coordinated surface Cu^2+^ ions. Furthermore, the documented enhancement in magnetization with decreasing CuO particle size has also been attributed to the increasing number of uncompensated surface spins contributing to the net moment [7,27].

## 2. Materials and Methods

The CuO nanoparticles employed in this study were prepared by a solvent deficient method [28,29]. In a typical synthesis, 53 g of Cu(NO_3_)_2_·2.5H_2_O and 38 g of NH_4_HCO_3_ were ground together in a mortar and pestle for approximately 1 min. Distilled H_2_O (10 mL) was then added to the mixture. The resulting precursor was rinsed with 0.5 L of distilled H_2_O before being calcined, in air, at 523 K (15 nm particles) for 1 h. Particles 8 and 25 nm in size were calcinated at 523 K and 623 K, respectively. The phase purities of all samples were confirmed by powder X-ray diffraction (PXRD) analyses performed with a PANalytical X’Pert Pro diffractometer (Malvern Panalytical Inc., Westborough, MA, USA) operating with Cu-*K*_α1_ radiation set at 45 kV and 40 mA (λ = 1.540598 Å). Data were acquired over the 2θ range of 10–90°. Only monoclinic CuO was observed. The average size of the particles was determined by powder X-ray diffraction by application of the Scherrer method [30]. The lattice parameters determined from Rietveld refinements of 15 nm CuO at 295 K (*a* = 4.6823 Å; *b* = 3.4242 Å; *c* = 5.1294 Å; β = 99.457°) are in good agreement with the literature data collected at room temperature for bulk and nanoscale CuO [3,22,23,24,25]. It should be noted that a range of values have been reported for the lattice parameters that may reflect variations in oxygen concentration of the samples [25]. This is important as oxygen vacancies have been shown to affect the magnetic properties of oxides containing 3*d* transition metals [8,9,31].

Variable temperature (7–400 K) Inelastic Neutron Scattering (INS) data were collected with the fine energy high resolution direct geometry chopper spectrometer SEQUOIA situated at the Spallation Neutron Source (SNS), Oak Ridge National Laboratory (ORNL) (Oak Ridge, TN, USA) [32,33]. INS spectra were collected at several incident energies (*E*_i_ = 10, 25 and 50 meV) that were selected by means of a Fermi chopper. Scattered neutrons of all energies were identified by position-sensitive detectors covering a wide range of scattering angles (−30° to +60° in the horizontal plane and ±18° in the vertical directions). Background spectra for an empty container were collected and subtracted from the sample data. The raw spectra were transformed from the time-of-flight and instrument coordinate bases to the dynamical structure factor S(Q,E) and finally to the S(Q,E)·E/[n(E,T) + 1] which directly relates to the intensity of magnons corrected for the population Bose factor [34,35].

## 3. Results and Discussion

Variable temperature spectra for 15 nm CuO nanoparticles are shown in Figure 1 and Figure 2. At 7 K, there is a faint dispersed signal at Q ≈ 0.84 Å^−1^ that dramatically increases in intensity as the temperature increases to 250 K (Figure 1). This dispersed signal originates from a purely magnetic Bragg peak with the scattering vector **Q** ≈ (0.5 0 −0.5) [36]. Figure 3 shows the evolution of the intensity of this elastic peak with temperature. The gradual reduction in the intensity of this peak indicates that the 15 nm CuO particles undergo a magnetic phase transition that commences at approximately 150 K and is fully complete by 225 K. This observed decrease in intensity is consistent with the commensurate → incommensurate antiferromagnetic transition at *T*_N2_.

In the low temperature (<*T*_N2_) commensurate phase of CuO, the antiferromagnetic spins are orientated parallel to [010] and are ordered along the wave vector **Q** = (0.5 0 −0.5) relative to the crystallographic basis of the atomic structure [23,24]. Conversion to the incommensurate antiferromagnetic phase that exists between *T*_N1_ and *T*_N2_ involves a 0.85° rotation of the antiferromagnetic ordering vector to (0.509 0 −0.483) [37,38]. In this incommensurate phase, the antiferromagnetic spins are helically modulated, and the magnetic moments rotate in a plane passing across the *b* axis and making an angle of approximately 74° with the ordering vector [39,40]. Propagation of magnons along these ordering vectors would be observed in the INS spectra at Q = 0.84 and 0.83 Å^−1^, respectively. Indeed, it is the excitation of spin precession waves in the direction of antiferromagnetic ordering vectors that give rise to the dispersed signal at Q ≈ 0.84 Å^−1^ in the CuO spectra shown in Figure 1 [25]. Unfortunately, as the directions of the ordering vectors in the commensurate and incommensurate are so similar, the expected shift in the Q-position of the magnons that would accompany the *T*_N2_ phase transition cannot be resolved in these data.

The degree of dispersion of the magnon signal is related to the strength of the exchange interactions between the magnetic spins [41]. The signals in the INS spectra of CuO show very large dispersion (almost vertical line at Q = 84 Å^−1^ in the Figure 1), which is consistent with the strong antiferromagnetic super-exchange interactions in this direction (*J_AFM_* = 67–80 meV) that are mediated by the oxygen orbitals of the Cu–O–Cu bridges [42,43,44,45]. As magnons are bosons, the observed increase in spin-wave intensity with increasing temperature up to about 100 K (Figure 1 and Appendix A) is a consequence of the number of magnons active within the CuO lattice being directly proportional to the Bose factor [41]. At *T* > 100 K, the intensity significantly increases and reaches maximum at about 225 K (close to *T*_N1_), and decreases at higher temperatures. The magnon excitations extend up to rather high energies (around 65–80 meV according to Ref. [39]), therefore our INS spectra show only their low energy part. The obtained results of increasing the magnon intensity (at *T* > 100 K) can be related to gradual softening of the dispersion curves of magnon spin-waves and decreasing the spin gap with temperature approaching *T*_N_ on heating (shifting of the magnons to lower energies), while at higher temperatures the short-range dynamic spin correlations exhibit slow dissipation.

Between 200 and 225 K, the CuO particles undergo a second magnetic phase transition (Figure 1 and Figure 3) and this third phase (paramagnetic) is retained up to at least 400 K. Chattopadhyay et al. published INS evidence of magnetic excitations along the scattering vectors **Q** = (1 0 1) and (1 0 −1) occurring in this phase of bulk CuO [46]. Yang et al. explained that observation of this signal above *T*_N_ is consistent with the large exchange antiferromagnetic constant [36]. Our data also show a weak broad magnon signal persisting at Q = 0.84 Å^−1^ in the spectra recorded at 300 K that becomes weaker with increasing temperature (Figure 2 and Figure 3).

INS spectra for CuO nanoparticles of different sizes are shown in Figure 4 (*T* = 183 K) and Figure 5 (*T* = 223 K). Consistent with the 7 K spectra for the 15 nm particles (Figure 1), magnons are very weak and almost invisible in the 7 K spectra for the 8 and 25 nm particles (see Appendix A). The intensity of dispersed magnon signal at Q ≈ 0.84 Å^−1^ in the spectra recorded at 183 and 223 K is particle-size dependent for both the commensurate (Figure 4) and incommensurate (Figure 5) antiferromagnetic phases (for large particles) and paramagnetic phase (for 8 nm particles), and increases in intensity with decreasing particle size. This “reverse finite size effect” has also been observed for magnons within antiferromagnetic α-Fe_2_O_3_ (hematite) [47]. The origin of this effect is not clear, but we postulate that it could be a consequence of the creation of single-domain particles at the nanoscale that facilitates the propagation of magnons through the magnetic lattice. Alternatively, inter-particle exchange interactions mediated through the increasing number of uncompensated surface spins inherent at the nanoscale (“superferromagnetism effect”) [48] could potentially enhance magnon propagation. This would be a result of the particles forming collective states with aligned magnetic moments that allow for collective magnon propagation [49] through multiple particles—in this effect, magnons are propagated through the inter-particle interface. Finally, it must be acknowledged that the role of lattice vacancies and their effect on the observations of this study are not fully known. Small particle sizes and lattice vacancies, for example, have been shown to affect the magnetic properties in both bulk and nanoscale CuO [7,27].

## 4. Conclusions

Newly-measured INS spectra of CuO nanoparticles provided important new insights into CuO nanoparticles. INS spectra revealed evidence of magnon propagation along the ordering vectors within the commensurate (*T* < *T*_N2_, **Q** = (0.5 0 −0.5)) and incommensurate (*T*_N2_ < *T* < *T*_N1_, **Q** = (0.506 0 −0.483)) antiferromagnetic phases of CuO. At low temperatures (*T* < 100 K), the intensity of the magnon signals increases with increasing temperature in accordance with the theory of the Bose population of states. Dynamic magnetic correlations are also clearly visible in INS spectra of the particles in the paramagnetic state (at *T* > *T*_N_). The intensity of the magnon signals in the INS spectra are found to be particle size dependent, and increase with decreasing particle size. The origin of this “reverse size effect” could be related to the formation of collective states in which the magnetic moments of closely positioned particles align and allow for the propagation of collective magnetic excitations. Future studies of interest should include a study of the structure and magnetic properties of nanoparticulate CuO under hydrostatic pressure to further elucidate the nature of its magnetic state (e.g., [50,51,52]).

## Figures and Tables

**Figure 1 nanomaterials-09-00312-f001:**
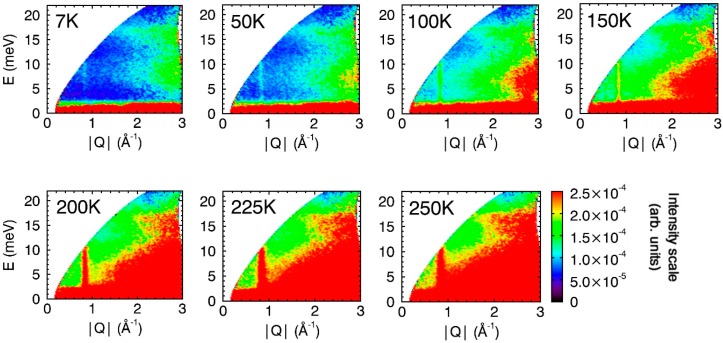
Variable temperature (7–250 K) INS spectra for 15 nm CuO particles. All Spectra were collected with *E*_i_ = 25 meV. To allow direct comparison, all spectra are drawn to the same intensity scale (bottom right). Additional spectra (for *T* = 200, 225, and 250 K) with different intensity scales are provided in the Appendix A.

**Figure 2 nanomaterials-09-00312-f002:**
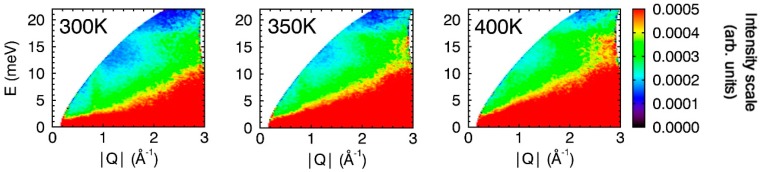
Variable temperature (300–400 K) INS spectra for 15 nm CuO particles. All Spectra were collected with *E*_i_ = 25 meV and are shown with the same intensity scale (right).

**Figure 3 nanomaterials-09-00312-f003:**
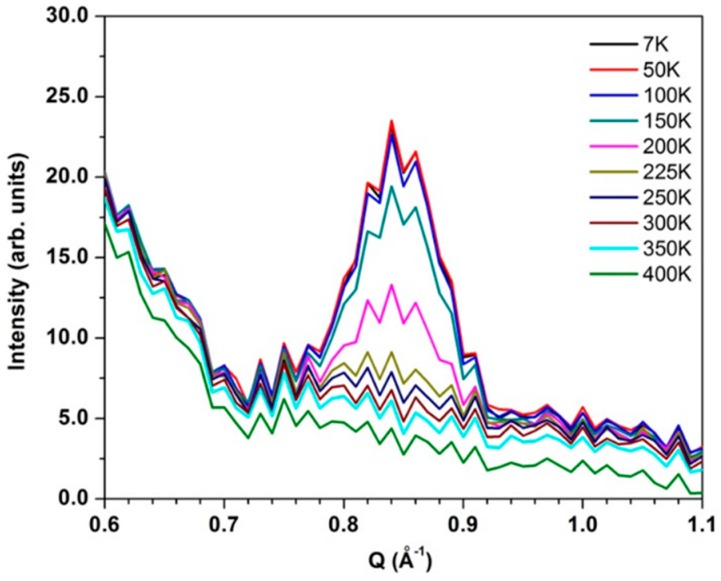
Temperature evolution of the magnetic Bragg peak (elastic signal, *E* = −0.5 to 0.5 meV) at around Q = 0.84 Å^−1^ in the INS spectra for 15 nm CuO particles.

**Figure 4 nanomaterials-09-00312-f004:**
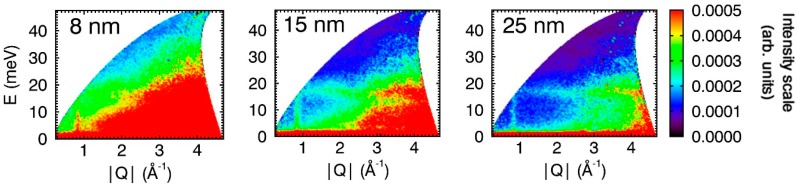
INS spectra for CuO nanoparticles of different sizes. All spectra were collected with *E*_i_ = 50 meV and at *T* = 183 K. At this temperature, the large particles are in the commensurate phase (<*T*_N2_), while the small 8 nm particles are in the paramagnetic state (*T*_N_ = 30 K and 50 K for the particles of size 5 and 10 nm, respectively [26]). The spectra are plotted with the same intensity scale (right).

**Figure 5 nanomaterials-09-00312-f005:**
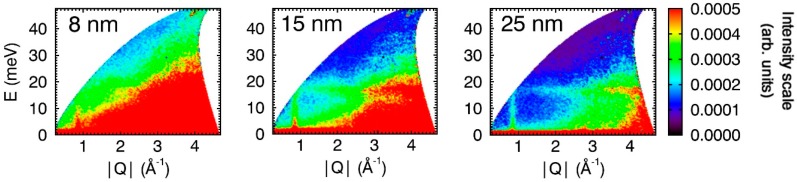
INS spectra for CuO nanoparticles of different sizes. All spectra were collected with *E*_i_ = 50 meV and at *T* = 223 K. At this temperature, the large particles are in the commensurate phase (<*T*_N2_), while the small 8 nm particles are in the paramagnetic state (*T*_N_ = 30 K and 50 K for the particles of size 5 and 10 nm, respectively [26]). The spectra are plotted with the same intensity scale (right).

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
