# Peer review of "New Insights about CuO Nanoparticles from Inelastic Neutron Scattering"

_nanomaterials, 2019, doi:10.3390/nano9030312_

Reviewer 1 Report

     Referee Report on paper “ New Insights about CuO Nanoparticles from Inelastic Neutron Scattering ” (nanomaterials-438493) by authors Elinor C. Spencer, Alexander I. Kolesnikov  and Nancy L. Ross submitted to Nanomaterials

This is interesting paper which contains useful information about the magnetic structure of CuO nanoparticles from inelastic neutron scattering. Evidences for the propagation of magnons in the directions of the ordering vectors of both the commensurate and helically modulated incommensurate antiferromagnetic phases of CuO had been presented. The temperature dependency of the magnon spin-wave intensity conformed to the Bose population of states at low temperatures at T≤100 K, then intensity significantly increased, with maximum at about 225 K and decreased at higher temperatures. Obtained results had been explained by gradual softening of the dispersion curves of magnon spin-waves and decreasing the spin gap with temperature approaching TN on heating, and slow dissipation of the short-range dynamic spin correlations at higher temperatures. Intensity of the magnon signal was particle size dependent, and increased with decreasing particle size. This “reverse size effect” had been explained by either creation of single domain particles at the nanoscale, or “superferromagnetism effect” and the formation of collective particle states. The reliability of the experimental results is beyond my doubt. I also have no serious objections to their interpretation. However, a few points should be improved. I think that this paper can be published only after corresponding corrections :

1.   The question of the oxygen nonstoichiometry of the samples obtained is very important. It is well known that the complex 3d-metal oxides easily allow the oxygen excess and/or deficit. A deviation from oxygen stoichiometry above a critical value leads to a change in intensity and, in the limit, to a change in the sign of exchange interactions :

(1). S.V. Trukhanov, I.O. Troyanchuk, A.V. Trukhanov, I.M. Fita, A.N. Vasil'ev, A. Maignan, H. Szymczak, Magnetic properties of La0.70Sr0.30MnO2.85 anion-deficient manganite under hydrostatic pressure, JETP Letters 83 (2006) 33-36. https://doi.org/10.1134/S0021364006010085.

(2). F. Gözüak, Y. Köseoğlu, A. Baykal, H. Kavas, Synthesis and characterization of CoxZn1−xFe2O4 magnetic nanoparticles via a PEG-assisted route, J. Magn. Magn. Mater. 321 (2009) 2170-2177. https://doi.org/10.1016/j.jmmm.2009.01.008.

This information should be mentioned in 3. Results and Discussion.

2.   Oxygen excess and deficit can increase and decrease the oxidation degree of 3d-metalls. The changing of charge state of 3d-metalls as a consequence of changing of oxygen content changes such magnetic parameters as total magnetic moment and Curie point. Moreover, oxygen vacancies effect on exchange interactions. Intensity of exchange interactions decreases with oxygen vacancy concentration increase. In complex oxides there is only indirect exchange. Exchange near the oxygen vacancies is negative according to Goodenough-Kanamori empirical rules. Oxygen vacancies should lead to the formation of a weak magnetic state such as spin glass :

(3). S.V. Trukhanov, L.S. Lobanovski, M.V. Bushinsky, V.V. Fedotova, I.O. Troyanchuk, A.V. Trukhanov, V.A. Ryzhov, H. Szymczak, R. Szymczak, M. Baran, Study of A-site ordered PrBaMn2O6-δ manganite properties depending on the treatment conditions, J. Phys.: Condens. Matter 17 (2005) 6495-6506. https://doi.org/10.1088/0953-8984/17/41/019.

This information should be discussed in 3. Results and Discussion.

3.   It is well known that for nanoscale magnetic compounds, surface tension is very important, which is expressed in the compression of the unit cell and an increase in the intensity of exchange interactions.

(4). M.A. Almessiere, Y. Slimani, H. Güngüne¸ A. Bayka, S.V. Trukhanov, A.V. Trukhanov, Manganese/Yttrium codoped strontium nanohexaferrites: evaluation of magnetic susceptibility and Mössbauer spectra, Nanomat. 9 (2019) 24-18. doi:10.3390/nano9010024.

Therefore, it is interesting to know the dependence of the unit cell parameter on the average particle size. This information should be also discussed in 3. Results and Discussion.

4.   Since competing effects, such as chemical composition deviation and surface tension, act in 3d-oxide nanoparticles, it is useful to study the structure and magnetic properties under the hydrostatic pressure to determine the nature of their magnetic state :

(5). S.V. Trukhanov, D.P. Kozlenko, A.V. Trukhanov, High hydrostatic pressure effect on magnetic state of anion-deficient La0.70Sr0.30MnOx perovskite manganites, J. Magn. Magn. Mater. 320 (2008) e88-e91. https://doi.org/10.1016/j.jmmm.2008.02.021.

(6). S.V. Trukhanov, A.V. Trukhanov, A.N. Vasiliev, H. Szymczak, Frustrated exchange interactions formation at low temperatures and high hydrostatic pressures in La0.70Sr0.30MnO2.85, JETP 111 (2010) 209-214. https://doi.org/10.1134/S106377611008008X.

This information should be also discussed in 3. Results and Discussion.

5.   The presented 6 papers should be inserted in References.

Author Response

We appreciate the thorough review and we have addressed the concerns raised. We focused on papers of previous studies of CuO and the Reviewer has broadened the discussion with inclusion from many other systems as highlighted below.  Nevertheless, the comments raise valid points that we have now addressed in the revised manuscript.

Comment 1: A further recognition of the importance of oxygen nonstoichiometry has been included in the introduction, results and discussion. Lattice parameters are included for comparison with other nano and bulk samples. We included the references:

F. Gözüak, Y. Köseoğlu, A. Baykal, H. Kavas, Synthesis and characterization of CoxZn1−xFe2O4 magnetic nanoparticles via a PEG-assisted route, J. Magn. Magn. Mater. 321 (2009) 2170-2177. https://doi.org/10.1016/j.jmmm.2009.01.008.

This information should be mentioned in 3. Results and Discussion.

Comment 2:  See above. And the following reference was included:

S.V. Trukhanov, L.S. Lobanovski, M.V. Bushinsky, V.V. Fedotova, I.O. Troyanchuk, A.V. Trukhanov, V.A. Ryzhov, H. Szymczak, R. Szymczak, M. Baran, Study of A-site ordered PrBaMn2O6-δ manganite properties depending on the treatment conditions, J. Phys.: Condens. Matter 17 (2005) 6495-6506. https://doi.org/10.1088/0953-8984/17/41/019.

Comment 3: Surface tension is acknowledged as important and the following reference is included.

M.A. Almessiere, Y. Slimani, H. Güngüne¸ A. Bayka, S.V. Trukhanov, A.V. Trukhanov, Manganese/Yttrium codoped strontium nanohexaferrites: evaluation of magnetic susceptibility and Mössbauer spectra, Nanomat. 9 (2019) 24-18. doi:10.3390/nano9010024.

Lattice parameters are included for comparison.

Comment 3: We agree that a high pressure study would be of intersest. We acknowledge this and reference previous studies of the effect og the structure and magnetic properties under the hydrostatic pressure to determine the nature of their magnetic state :

S.V. Trukhanov, I.O. Troyanchuk, A.V. Trukhanov, I.M. Fita, A.N. Vasil'ev, A. Maignan, H. Szymczak, Magnetic properties of La0.70Sr0.30MnO2.85 anion-deficient manganite under hydrostatic pressure, JETP Letters 83 (2006) 33-36.

S.V. Trukhanov, D.P. Kozlenko, A.V. Trukhanov, High hydrostatic pressure effect on magnetic state of anion-deficient La0.70Sr0.30MnOx perovskite manganites, J. Magn. Magn. Mater. 320 (2008) e88-e91.

S.V. Trukhanov, A.V. Trukhanov, A.N. Vasiliev, H. Szymczak, Frustrated exchange interactions formation at low temperatures and high hydrostatic pressures in La0.70Sr0.30MnO2.85, JETP 111 (2010) 209-214.

Reviewer 2 Report

The article is devoted to inelastic neutron scattering (INS) measurements on cupric oxide nanoparticles. CuO nanoparticles seem to be well-investigated, and one can find a lot of information about their magnetic properties in the literature. But the authors indeed delivered some novel results in the article considered. The magnons propagation in antiferromagnetic phases of CuO has been considered in details. Especially “reverse size effect” seems to be particularly interesting. This effect could be related to the formation of collective states in which the magnetic  moments of closely positioned particles align and allow for the propagation of collective magnetic excitations. In my opinion this phenomenon is worth of further explanation, but within the framework of another article.

The article is well-written, and seems to be worth of publication. Nevertheless some minor corrections are necessary. They are listed below.

Line 67: calcination in 250K (-23 deg. C) seems to be impossible. I think, that authors means degrees centigrade not Kelvins, or I do not understood something. This should be corrected, or explained to me.

Line 73: the INS abbreviation is not explained. I think, that it should be Inelastic Neutron Scattering (INS) uniformly, like previous research description.

Line 92: I think, that it is some auxiliary text, that should be delated.

Figure 1: the pictures quality is very low, scale is unreadable. Axes description should be corrected (fonts should be enlarged), and the figures should be provided in some vectorial format (for redaction)

Figure 2: similar, fonts should be enlarged.

Author Response

We are delighted that the reviewer acknowledges that our manuscript is well-written  and worthy of publication.  The minor corrections listed below have been made: 

Line 67: calcination in 250K (-23 deg. C) seems to be impossible. I think, that authors means degrees centigrade not Kelvins, or I do not understood something. This should be corrected, or explained to me.

Thank you for catching this!

Line 73: the INS abbreviation is not explained. I think, that it should be Inelastic Neutron Scattering (INS) uniformly, like previous research description.

Done.

Line 92: I think, that it is some auxiliary text, that should be delated.

Done.

Figure 1: the pictures quality is very low, scale is unreadable. Axes description should be corrected (fonts should be enlarged), and the figures should be provided in some vectorial format (for redaction)

Figure 2: similar, fonts should be enlarged.

Both Figures have been revised. 

Reviewer 3 Report

Paper can be aaccepted after the minor correction: conclusions should be stated in more quantitative way.

Author Response

We are delighted of the positive review.

We have revised the paper including making more quantitative conclusions.

Round  2

Reviewer 1 Report

Referee Report on paper “ New Insights about CuO Nanoparticles from Inelastic Neutron Scattering ” (nanomaterials-438493_v2) by authors Elinor C. Spencer, Alexander I. Kolesnikov  and Nancy L. Ross submitted to Nanomaterials

The article was quite satisfactorily corrected and improved, so I see no reason not to publish it.
